# Weakening of the South Asian summer monsoon linked to interhemispheric ice-sheet growth since 12 Ma

Zhengquan Yao [1,2] ✉, Xuefa Shi [1,2] ✉, Zhengtang Guo[3,4,5], Xinzhou Li [6,7], B. Nagender Nath[8], Christian Betzler [9], Hui Zhang[1,2], Sebastian Lindhorst[9] & Pavan Miriyala[10]

The evolution and driving mechanism of the South Asian summer monsoon (SASM) are still poorly understood. We here present a 12-Myr long SASM record by analyzing the strontium and neodymium isotopic composition of detrital components at IODP Exp. 359 Site U1467 from the northern Indian Ocean. The provenance investigation demonstrates that more dust enriched in εNd from northeastern Africa and the Arabian Peninsula was transported to the study site by monsoonal and Shamal winds during the summer monsoon season. A two-step weakening of the SASM wind since ~12 Ma is proposed based on the εNd record. This observational phenomenon is supported by climate modeling results, demonstrating that the SASM evolution was mainly controlled by variations in the gradient between the Mascarene High and the Indian Low, associated with meridional shifts of the Hadley Cell and the Intertropical Convergence Zone, which were caused by interhemispheric ice-sheet growth since the Middle Miocene.

The South Asian monsoon (SAM), an important component of the global climate system[1], impacts the economic and wellbeing activity of nearly half of the world's population. Consequently, its long-term evolution and the underlying causes have attracted much attention from the paleoclimate research community[2–7]. Increasing evidence indicates that the Asian monsoon was initiated no later than the Early Miocene[3,5,7]; however, the mechanisms driving its subsequent evolution are highly debated. Suggested mechanisms include the development of high topographic relief linked to the uplift of the Tibetan-Himalayan system[8,9], global cooling as a consequence of polar ice-sheet development[10,11], the retreat of the Tethys Sea[12], and variations in $p\mathrm{CO_2}$[5,13]. Among these, much attention has been paid to the global cooling associated with the expansion of polar ice-sheets, which is

evidenced by both sedimentary records[7,10,11] and climate modeling[14]. However, details of exactly how global cooling controlled the Asian monsoon remain unclear. In contrast to the long-term evolution of the East Asian monsoon (EAM), which is recorded by many eolian sequences in northern China[3,9,15], SAM records extending to the Middle Miocene are scarce, especially from within the marine realm of the SAM-influenced region, with few exceptions[6,7,16].

To fill this knowledge gap, we measured the strontium and neodymium isotopic compositions of detrital fraction in a 708-m long sedimentary sequence from IODP expedition 359 Site U1467 (4° 51′ N, 73° 17′ E; ~12 Myr[7]), recovered from the Maldives Inner Sea in the southeastern Arabian Sea, from a water depth of ~487 m (Fig. 1a, b). Linking sedimentary provenance and Nd isotopic composition with

[1]Key Laboratory of Marine Geology and Metallogeny, First Institute of Oceanography, Ministry of Natural Resources, Qingdao, China. [2]Laboratory for Marine Geology, Qingdao National Laboratory for Marine Science and Technology, Qingdao, China. [3]Key Laboratory of Cenozoic Geology and Environment, Institute of Geology and Geophysics, Chinese Academy of Sciences, Beijing, China. [4]CAS Center for Excellence in Life and Paleoenvironment, Beijing, China. [5]University of Chinese Academy of Sciences, Beijing, China. [6]State Key Laboratory of Loess and Quaternary Geology, Institute of Earth Environment, Chinese Academy of Sciences, Xi'an, China. [7]CAS Center for Excellence in Quaternary Science and Global Change, Xi'an, China. [8]Geological Oceanography Division, CSIR-National Institute of Oceanography, Dona Paula, Goa, India. [9]Institute of Geology, CEN, University of Hamburg, Hamburg, Germany. [10]CSIR-National Geophysical Research Institute, Hyderabad, India. ✉e-mail: yaozq@fio.org.cn; xfshi@fio.org.cn

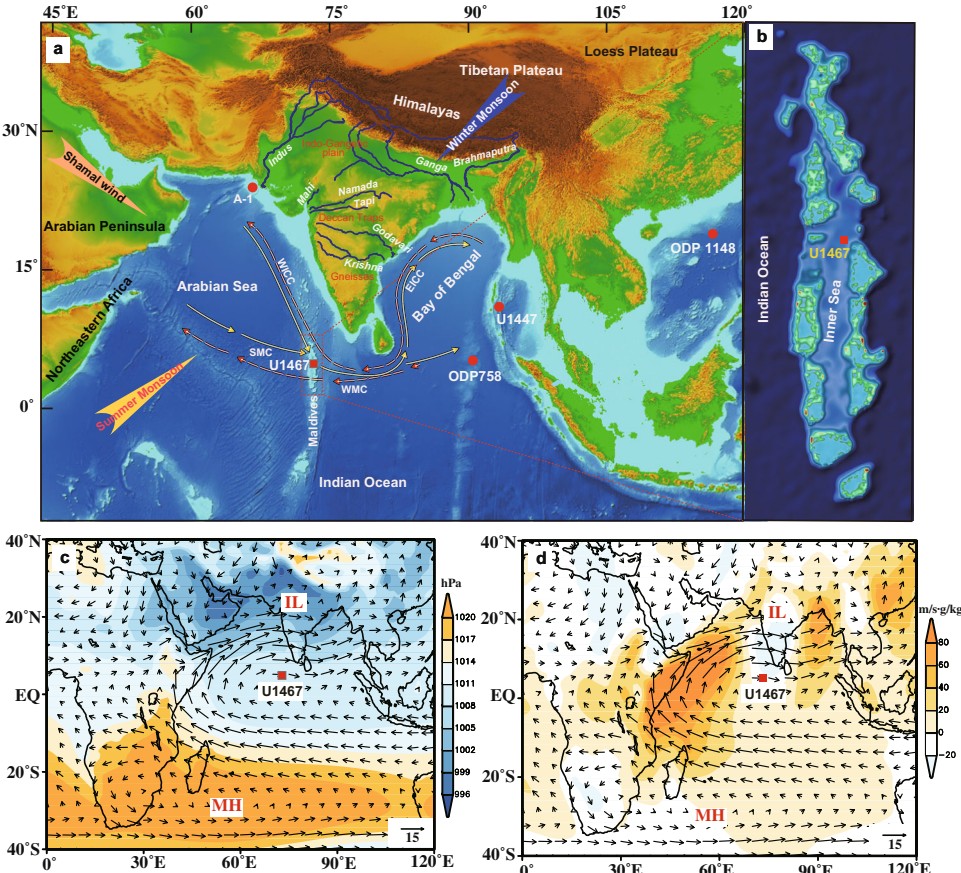

**Fig. 1 | General information about the study area. a** Bathymetric map showing the location of IODP Exp. 359 Site U1467 and other sites mentioned in the text. The circulations (modified after ref. [55]) in the northern Indian Ocean in summer (thick yellow line) and winter (dashed pink line) are indicated, including the East India Coastal Current (EICC), West India Coastal Current (WICC), Summer Monsoon Current (SMC) and the Winter Monsoon Current (WMC). **b** Map showing the northern part of the Maldives. **c, d** Model-exported sea-level pressure (units: hPa), meridional vapor flux (units: m/s·g/kg) and wind vector at 850 hPa during the present summer season (June–July–August; JJA). The Mascarene High (MH; 40°–90°E, 25°–35°S) and the Indian Low (IL; 40°–80°E, 20°–30°N) are labeled. The base maps (1a, b) are generated using open access software SimpleDEMViewer (http://www.jizoh.jp) with data from https://www.ngdc.noaa.gov/mgg/global/global.html.

monsoonal climate, we reveal a two-step decline of the SASM over the past ~12 Myr. We suggest that this evolutionary pattern was closely related to the meridional shifts of the Hadley Cell caused by the growth of interhemispheric ice sheets since the Middle Miocene, which is supported by climate modeling.

## Results and discussion

### Sr and Nd isotopes and sediment provenance discrimination

The Sr and Nd isotopic compositions in the silicate phases of marine sediments provide a means of tracing their sources and variations, and are therefore commonly used as proxies for sediment provenance[17–19]. In addition to provenance changes related to the parent rocks, Sr isotopes tend to be influenced by grain-size effects and the degree of weathering[20–22], whereas the Nd isotopic ratio is considered to be less altered during transportation and/or after deposition[21]. At Site U1467, the Sr and Nd isotopic compositions of the detrital fraction were likely primarily controlled by changes in sediment provenance, rather than by possible changes in the degree of chemical weathering and/or grain-size effects based on a rough anti-correlation between the Sr and Nd isotopic compositions in the sediments (Supplementary Fig. 1c, d). Moreover, the Sr and Nd isotopic compositions of terrigenous sediments in the northern Indian Ocean have been confirmed to mainly reflect sediment source, and significant influences of grain size and/or chemical weathering on the Sr isotopic values are excluded[17,18,23].

In contrast to other open ocean settings, the Maldives in the southeastern Arabian Sea is unique, in comprising a double row of atolls that encloses an isolated perched basin (Fig. 1b). It is elevated between several thousand meters above the surrounding seafloor, and therefore shielded effectively from terrestrial input[7], such that fluvial materials get transported by currents to the Maldives are very limited[24]. Therefore, the Maldives Inner Sea acts as a natural sediment trap best suited for dust deposition[7]. Evidence of sedimentology, geochemistry and morphology of sediment grains from Site U1467 all suggest that the lithogenic fraction is primarily dominated by eolian materials[25,26]. Consequently, the lithogenic fraction at Site U1467 is of eolian origin, and the contribution of fluvially-derived material supplied by currents from the Arabian Sea and the Bay of Bengal can be neglected[24–26]. Previous studies have demonstrated that the input of eolian dust to the Arabian Sea is primarily linked to the southwesterly monsoonal winds and subordinated northwesterly Shamal winds, both during the summer monsoon season[26–29]. These winds can transport large amounts of dust to the eastern Arabian Sea, and even to the northeastern Indian Ocean, where it is scavenged by summer monsoonal precipitation and wet-deposited[26–29]. On the contrary, material from the Indian subcontinents, mainly the Thar desert, could also be transported to the site location by winter monsoonal winds during the winter season[30]. The Arabian desert dust reveals a more radiogenic εNd of −6, and ⁸⁷Sr/⁸⁶Sr values between 0.707 and 0.715[29]. The Sr and Nd isotopic composition of a Holocene eolian sequence from the western Arabian

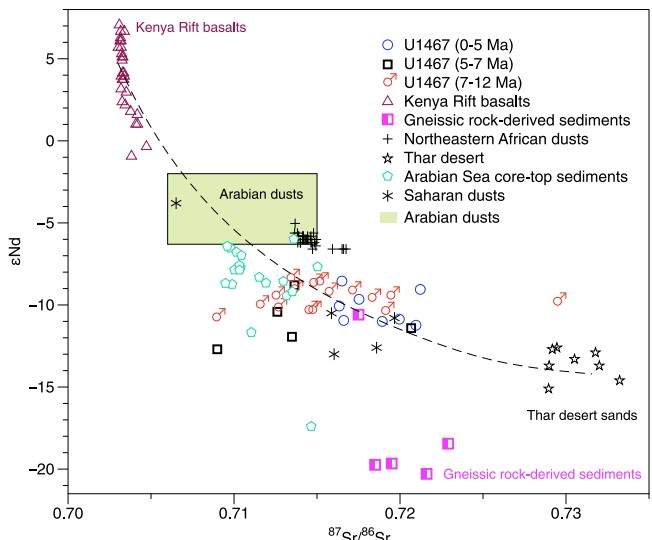

**Fig. 2 | Sediment provenance discrimination of Site U1467.** $^{87}$Sr/$^{86}$Sr isotopic ratios versus εNd diagram for Site U1467 along with the potential sources of northeastern African rift[34], gneissic rock-derived sediments in the southern Indian continent[19], Saharan dusts[32], northeastern African dusts[31], Indian Thar desert[36], Arabian dusts and the core-top sediments from the Arabian Sea[29].

Sea off Somalia showed average values of 0.714 and −6.03, respectively[31]. The Saharan desert materials have relatively less radiogenic Nd with an average value of −10.1 (Sr: 0.712 to 0.720)[32]. The average εNd value of marine sediments and rocks from the margins of the northeastern African and the southern Arabian Peninsula too are highly radiogenic (average −4.6)[33]. These more radiogenic Nd values observed in the northeastern Africa and the Arabian Peninsula are consistent with the Kenya Rift basalts (northeastern Africa, εNd: −0.9 to 7.1)[34] and the Arabian-Nubian shield (εNd: 1 to 6)[35]. By contrast, the Sr and Nd isotopic composition of the Indian Thar deserts showed much less radiogenic Nd values (Nd: −15.1 to −12.6; Sr: 0.729 to 0.733)[36].

The Sr and Nd isotopic compositions of the sediments at Site U1467 since ~12 Ma align along a hyperbolic mixing curve linking materials from northeastern Africa and the Arabian Peninsula, with more radiogenic εNd[31–34], and from the Thar desert, with less radiogenic εNd values[36] (Fig. 2). These results indicate that the sediments deposited during the last ~12 Myr are of mixed provenance, with a major contribution from northeastern Africa/Arabian Peninsula and a minor contribution from the Thar desert sediments. However, three samples with the least radiogenic εNd for the interval of ~7−5 Ma deviate from this mixing line (Fig. 2), suggesting the possible influence of additional sediment sources. The southern Indian continent is dominated by gneissic rocks, and gneissic rock-derived sediments display $^{87}$Sr/$^{86}$Sr and εNd values ranging from 0.7175 to 0.7229 and from −20 to −18, respectively[19]. Thus the sediments in this area could be an additional source for the sediments aged ~7−5 Ma at Site U1467. Our predicted provenance change at Site U1467 is further supported by the Nd isotope of core-top sediments from the Arabian Sea and its surrounding margins[29,33]. Spatial distribution of εNd values of the detrital fraction from these areas reveals clear progressively decreasing trend from the northeastern Africa/Arabian Peninsula to the western and eastern Arabian Sea[29,33] (Supplementary Fig. 2).

It has been highlighted that the Nd cycle in the ocean is complicated and could be influenced by factors such as fluvial input, boundary exchange, and ocean-circulation borne Nd[37–40]. Boundary exchange has been used mainly to explain the effect of sediments on dissolved Nd in seawater and not vice versa[41], since the Nd concentration in sediments is 2−3 orders of magnitude higher than in dissolved seawater. Moreover, the lithic grains in the study area were primarily of eolian origin with

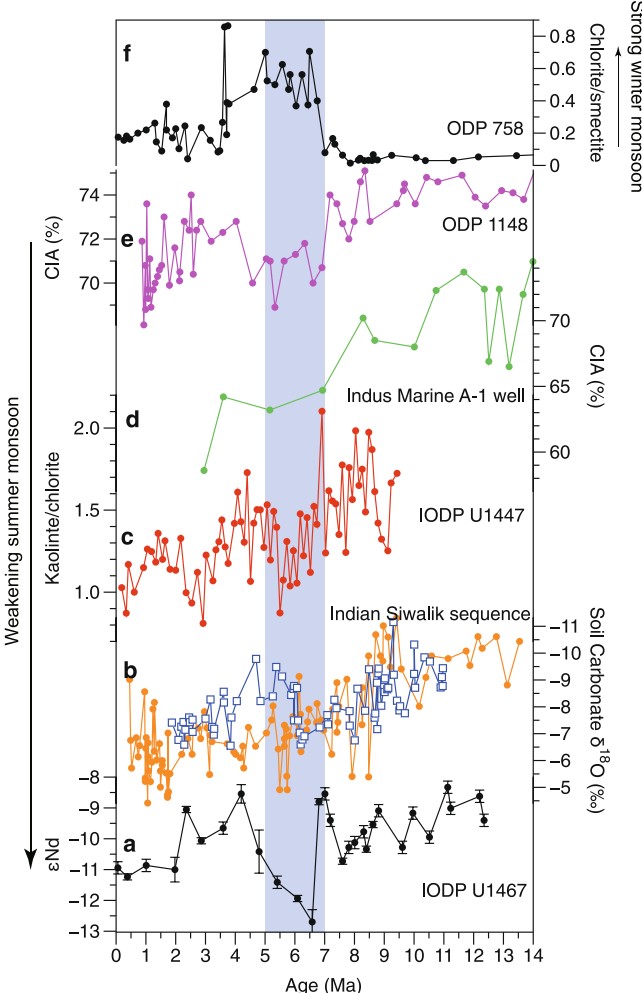

**Fig. 3 | Comparisons of the South Asian summer monsoon (SASM) proxies with clay mineral data from ODP Site 758 during the last ~12 Myr. a** Variations in εNd (2sigma S.D.) at Site U1467 (this study). **b** δ$^{18}$O of soil carbonate from the Indian Siwalik formation[46,47]. **c** Kaolinite/chlorite ratio from IODP Site U1447 in the western Andaman Sea[50]. **d** Chemical weathering proxies (CIA) from the Indus Marine A-1 well in the northern Arabian Sea[48]. **e** CIA index from ODP Site 1148 in the northern South China Sea[49]. **f** Chlorite/ smectite ratio of ODP Site 758[54]. The vertical shaded area represents the time interval of ~7−5 Ma.

relatively slow accumulation rates, and thus were less influenced by the boundary exchange which is often significant in the continental margins with high sediments supply[37,41]. Therefore, we could preclude a significant influence of these processes on detrital Nd values at Site U1467. Although the Antarctic Intermediate Water (AAIW) has potentially influenced the intermediate water of the northern Indian Ocean[42], but the εNd of AAIW is generally fluctuated between −8.4 and −6 for the glacial and interglacial periods[43,44], which is far more radiogenic than those of our lithogenic fraction sediment data (Fig. 3a). Further, our detrital Nd record is quite different from the seawater Nd pattern of intermediate waters of ODP Sites 707 (1552 m) and 775 (1661 m) in the northern Indian Ocean[45] (Fig. 4a). All the above evidences suggest that the authigenic Nd originated from the AAIW has no pronounced influence on the εNd of the detrital fraction at Site U1467.

The εNd values of the detrital components at Site U1467 display a two-step decrease starting from ~12 Ma and at ~4 Ma, respectively, interrupted by an abrupt decrease at ~7−5 Ma (Fig. 3a). The εNd values range from −10.7 to −8.3 (average −9.5; Fig. 3a, Supplementary Table 1) during the interval from ~12 Ma to −7 Ma. During the interval of ~7−5 Ma, εNd decreased to a mean value of −11.1 (Fig. 3a, Supplementary

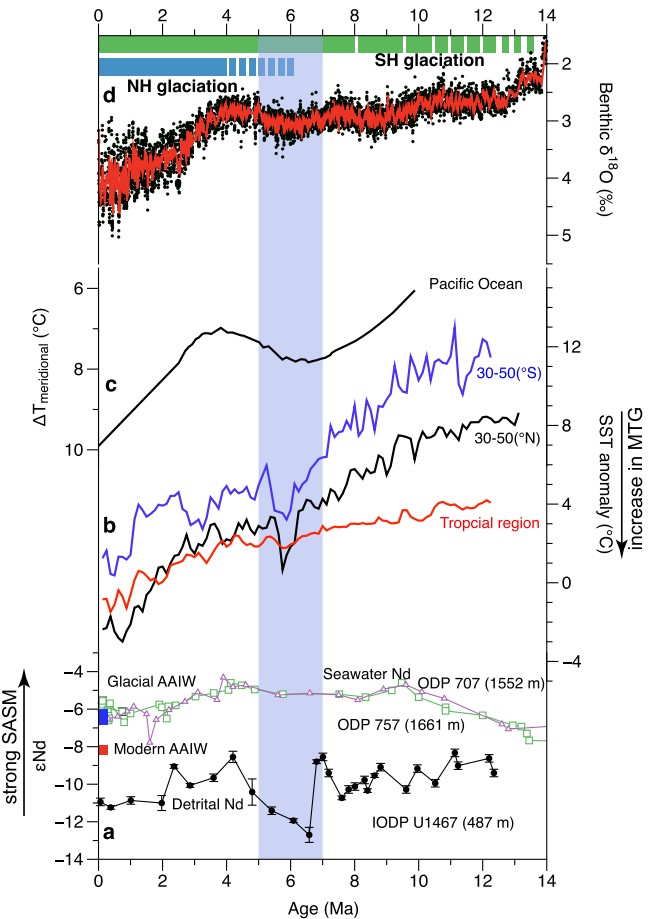

**Fig. 4 | Comparisons of the South Asian summer monsoon (SASM) evolution with other climatic records. a** Comparison of detrital εNd (2sigma S.D.) at Site U1467 with seawater εNd at ODP Sites 757 and 707 in the northern Indian Ocean[45]. The εNd ranges of the glacial and modern Antarctic Intermediate Water (AAIW)[43,44] are indicated by blue and red rectangles. **b** Stacks of alkenone $U^k_{37}$-based sea-surface temperature (SST) anomalies relative to today for subtropical (30–50° N, 30–50° S) and tropical regions[62], with lower values indicating strong meridional temperature gradients (MTG). **c** Meridional temperature gradients of the Pacific calculated between sites from the northern subtropical and equator region[61]. **d** Benthic oxygen isotope record from ODP Site 1146[11] (21-point moving average) along with blue and green bars denoting the generally accepted chronology of the Southern and Northern Hemisphere Cenozoic glaciation[60,64–67]. The vertical shaded area represents the time interval of ~7–5 Ma.

Table 1). Thereafter, the values increased from a minimum of −12.7 to −8.5 at ~ 4 Ma, followed by a more pronounced decrease from −8.5 to −11.2 since ~4 Ma (Fig. 3a; Supplementary Table 1).

**South Asian summer monsoon evolution over the past 12 Myr**
The Nd isotopic composition of the sediments at Site U1467 can thus serve as a proxy for the relative intensity of the SASM, with more radiogenic εNd values indicating an increased dust contribution to the study area from northeastern Africa and the Arabian Peninsula by enhanced monsoonal and Shamal winds, and concurrently increased precipitation. Based on the Nd isotopic data for Site U1467, the contribution of the dust from northeastern Africa and the Arabian Peninsula decreased after ~12 Ma. This decrease in the dust supply can be attributed to changes in the SASM winds, which shows a two-step weakening beginning at ~12 Ma and ~4 Ma, interrupted by an abrupt decrease at ~7–5 Ma (Fig. 3a). This evolutionary pattern of the SASM, inferred from the Nd isotopes at Site U1467, is largely consistent with the results of previous studies[6,46–50].

Oxygen isotope ratios of pedogenic carbonate depend upon the isotopic ratio of the soil water derived from the local precipitation. Consequently, the $\delta^{18}O$ of pedogenic carbonate can be used to decipher the average isotopic composition of precipitation and thus the summer monsoon rainfall[46,47]. Oxygen isotope data from soil carbonate nodules from the Indian Siwalik sequences show a weakening of the SASM since ~12 Ma, with a slight decrease in intensity at ~6–5 Ma[46,47] (Fig. 3b), which is largely consistent with the data presented herein. Continental weathering in the monsoonal region, including chemical weathering and physical erosion, is usually linked to the monsoon climate[22,48]. A higher degree of chemical weathering and high sediment flux usually indicate a wetter/warmer climate, and thus a stronger SASM. Proxies of chemical weathering intensity, namely the chemical index of alteration (CIA) from the Indus Fan, indicate a strong SASM during ~16–10 Ma interval, followed by a gradual weakening during ~10–3 Ma[48] (Fig. 3d). The CIA index at the ODP Site 1148 from the northern South China Sea also reveals similar monsoonal evolutional pattern with our Nd records[49] (Fig. 3e). Changes in the kaolinite/chlorite ratio at Site U1447 in the western Andaman Sea, a proxy of chemical weathering in warm humid climate relative to the physical erosion under cold and dry climatic conditions[51], demonstrate a weakening of the SASM since the Late Miocene (~10 Ma), interrupted by a further decrease at ~5–7 Ma[50] (Fig. 3c). A compilation of sediment fluxes inferred from seismic data from the Asian marginal seas shows peak values in the Early-Middle Miocene (24–11 Ma), followed by a decreasing sediment accumulation rate with a minimum at ~8–6 Ma, which is interpreted as reflecting reduced erosion due to a weak summer monsoonal climate[48,52]. The coherence of these records, originating from different locations and based on different proxies, provides robust evidence for the two-step weakening of the SASM over the last ~12 Myr.

In addition to the influence of the weakened SASM which reduced the dust supply with more radiogenic Nd, the substantial decrease in the εNd values during the interval of ~7–5 Ma could be due to addition of sediment material with the least radiogenic εNd values (−20 to −18), possibly from the southern Indian continent[19], to the core site (Fig. 2). These materials were possibly transported by the strengthened winter monsoonal winds, as previous studies have argued for a cold-drying climate and intensified winter monsoon during this time interval[11,48,53]. Our interpretation is consistent with a two-fold increase in sediment accumulation rate at Site U1467 during the interval of ~7–5 Ma[7] (Supplementary Fig. 1a). This inference is also supported by an abrupt increase in the ratio of chlorite/smectite from ODP Site 758 in the northeastern Indian Ocean (Fig. 3f), which implies a higher contribution of materials derived from the physical erosion of highlands of the Himalayan river basins[54], most likely transported by the East India Coastal Current[55] under strong winter monsoonal wind.

**South Asian summer monsoon evolution linked to interhemispheric ice-sheet growth**
The South Asian summer monsoon is driven by the pressure gradient between the Indian Low (IL) over the Asian continent and the Mascarene High (MH) in the southern Indian ocean[56,57] (Fig. 1c, d). The MH is forced by the descending limb of the Hadley Cell in the Southern Hemisphere, determined by Indian Ocean surface temperatures[58], which intensifies during the boreal summer. Simultaneously, IL develops driven by direct sensible heating of the Asian continental landmass[59]. Consequently, the variations in the IL and the MH are related to the temperature of the Northern and Southern Hemispheres, both of which are closely linked to the ice-sheet development on geological timescales. As expected, the reconstructed weakening of the SASM is largely synchronous with global cooling[11,60] (Fig. 4d). Specifically, it evolved synchronously with the increase in meridional temperature gradients in the northern Pacific[61] (Fig. 4c), and over a large area globally, as indicated by stacks of alkenone $U^k_{37}$-based sea-surface temperature anomalies in the tropics and the subtropical

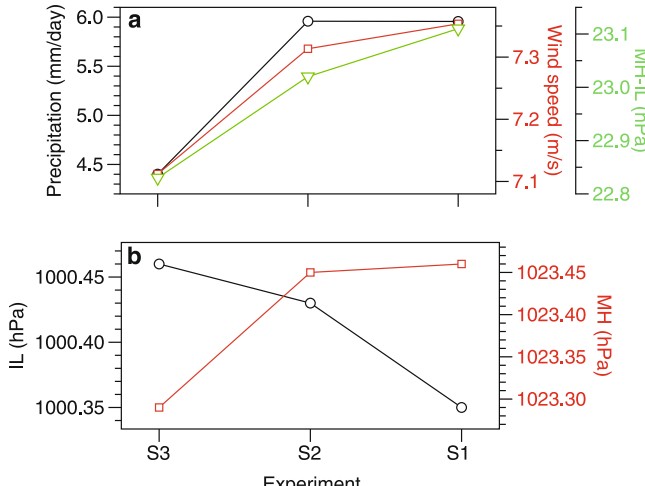

**Fig. 5 | Variations in model-exported parameters. a** Simulated 850-hPa wind speed (m/s) and precipitation (mm/day) averaged for the summer season (June, July and August; JJA) in the area (50°–75°E, 5°S–20°N), pressure gradient (MH-IL) of the Mascarene High (MH; 40°–90°E, 25°–35°S) relative to the Indian Low (IL; 40°–80°E, 20°–30°N) under three scenarios with growth of the Antarctic and Greenland ice-sheet. **b** Variations in MH and IL under three scenarios. Scenarios of S1 and S2 represent the ice cover accumulated in the East Antarctica and the whole Antarctica, respectively. Scenario S3 represents the pre-industrial ice cover in Greenland superimposed on a constant ice-sheet cover in the whole Antarctica.

region of both hemispheres[62] (Fig. 4b). We suggest that the changes in the MH and IL pressure system, meridional temperature gradients, and the polar ice-sheet growth were linked coherently, which generated the two-step decline of the SASM since ~12 Ma.

The proposed linkage between the SASM evolution and the growth of polar ice sheets over the last ~12 Myr was examined using numerical simulations. Sensitivity results reveal relatively consistent variations in the MH-IL pressure gradient, 850-hPa wind speed, and precipitation during the summer season (June-July-August; JJA) in the study area (Fig. 5a), which are closely related to the SASM intensity. These three atmospheric parameters firstly decreased, except that the precipitation slightly increased, with expanded ice cover in the West Antarctica, further decreased with increase in the volume of the Greenland ice sheet (Figs. 5a and 6a, b).

As the Intertropical Convergence Zone (ITCZ) is associated with the ascending branch of the Hadley Cell, its position is closely coupled with the movement of the Hadley Cell, as well as the MH and IL. A previous study suggests that the ITCZ, and thus the Hadley Cell, was displaced toward the warmer hemisphere[63]. There was a major expansion of the East Antarctic ice sheet from ~13.9 Ma onwards and a further expansion of the West Antarctic ice sheet until ~8 Ma[60,64,65] (Fig. 4d) leading to faster cooling, and consequently to enhanced meridional temperature gradients, in the Southern Hemisphere relative to the Northern Hemisphere[62] (Fig. 4b). This change would have driven the Hadley Cell of both hemispheres northward and caused a deviation of the subsiding branch of the Southern Hemisphere Hadley Cell away from the previous position of the MH, while at the same time moving the ascending branch of the Hadley Cell out of the previous position of the IL. Consequently, during the boreal summer the MH and IL weakened (Fig. 5b; Fig. 6c), leading to the decline of the SASM strength from ~12 Ma to ~6 Ma (Fig. 2). The growth of the Greenland ice sheet, however, operated in the opposite way. The development of ephemeral Northern Hemisphere ice sheet between ~6 and 8 Ma[60,66], followed by the significant expansion of the Greenland ice sheet commenced at ~4 Ma[60,66,67] (Fig. 4d), causing a cooling in the mid-latitude Northern Hemisphere and enhanced meridional temperature gradients[62] (Fig. 4b). This pushes the Hadley Cell back southward,

possibly leading to the subsiding and ascending branches of the Hadley Cell deviated from the MH and IL in the opposite direction. This change would also weaken the MH and the IL during the boreal summer (Figs. 5b and 6d), leading to a pronounced decline of the SASM since ~4 Ma (Fig. 2). Our predicted movement of the Hadley Cell is consistent with the evolutionary history of the ITCZ inferred both from model simulations and proxy reconstructions. During the Middle to Late Miocene (~12 to ~6 Ma), the ITCZ moved northward[68–70], to a location farther north than its present position, until the early Pliocene[71,72]. Subsequently, there was a southward shift of the ITCZ from ~5 Ma to ~4 Ma[72], followed by a further southward shift towards the equator after ~4 Ma[73]. This model-generated pattern is consistent with the proxy-reconstructed SASM, supporting our inference that interhemispheric ice-sheet development may have played a key role in regulating the evolution of the SASM.

In addition to global cooling, the uplift of the Himalayan-Tibetan region[3,8], closure of the Tethys Sea[12], and changes in $p\mathrm{CO_2}$[5,13] may also have contributed to the evolution of the SASM. However, the continental collision between India and Eurasia is thought to have commenced no later than 40 Ma[74]. The Himalayan-Tibetan region may have reached its present height by the Early-Middle Miocene[75] and there was no significant tectonic uplift thereafter. Similarly, the final closure of the Tethys Sea was much earlier (by ~15 Ma)[76] which precludes its potential impact on changes in the SASM. There is uncertainty regarding the role of $p\mathrm{CO_2}$ in affecting monsoon climate[77]. However, a recent study excluded it as a controlling factor in the evolution of the East Asian monsoon[78]. Considering that $p\mathrm{CO_2}$ and climate are coupled throughout the Cenozoic[79], we suggest that $CO_2$ might not directly drive the evolution of the SASM, but might act through its modulation of glaciations of the polar regions[80]. In summary, our proxy records combined with numerical modeling, demonstrate for the first time a linkage between the SASM and interhemispheric ice-sheet growth via the modulation of the meridional shifts of the Hadley Cell and its associated MH-IL pressure gradient since the Middle Miocene. Our study thus provides a more comprehensive interpretation of how interhemispheric ice-sheet development and associated global atmospheric circulation regulated SASM evolution on geological timescales.

## Methods
### Core description and chronology
IODP Exp. 359 Site U1467 (4° 51′ N, 73° 17′ E, water depth 487 m) in the Maldives Inner Sea was cored through a sequence of carbonate drift deposits (Fig. 1). The recovered 708-m thick sequence is dominated by unlithified and partially lithified foraminifer-rich, fine-grained silty carbonates[81], with carbonate content ranging from 80% to 95%[82] (Supplementary Fig. 1b). The chronology of Site U1467 was established using biostratigraphy, based on calcareous nannofossil and planktonic foraminifers[7] (Supplementary Fig. 1a). According to the age model, the 708-m sequence provides a continuous record spanning the last ~12 Myr, with an average linear sedimentation rate of ~5.0 cm/kyr[81].

### Sr-Nd isotope analysis
The strontium and neodymium isotopes of 29 samples from Site U1467 were analyzed using a High-Resolution Multi-Collector Inductively Coupled Plasma Mass Spectrometer (HR-MC-ICPMS). Pretreatment methods followed ref. [83]. Briefly, the samples were treated with 10% acetic acid and 5% hydrogen peroxide to remove carbonates and organic matter, respectively. Subsequently, 1 M hydroxylamine hydrochloride in 25% acetic acid was added to remove the Fe−Mn oxide fraction and potential residual carbonate. The residue was then transferred to a Teflon bottle and completely dissolved using an HF−HNO₃−HClO₄ mixture. Sr and Nd were extracted from the solution using standard ion-exchange procedures. The Sr and Nd isotopic ratios obtained were normalized to $^{86}\mathrm{Sr}/^{88}\mathrm{Sr}$ (0.1194) and $^{146}\mathrm{Nd}/^{144}\mathrm{Nd}$ (0.7219), respectively. Repeated analyses of the NBS987 standard

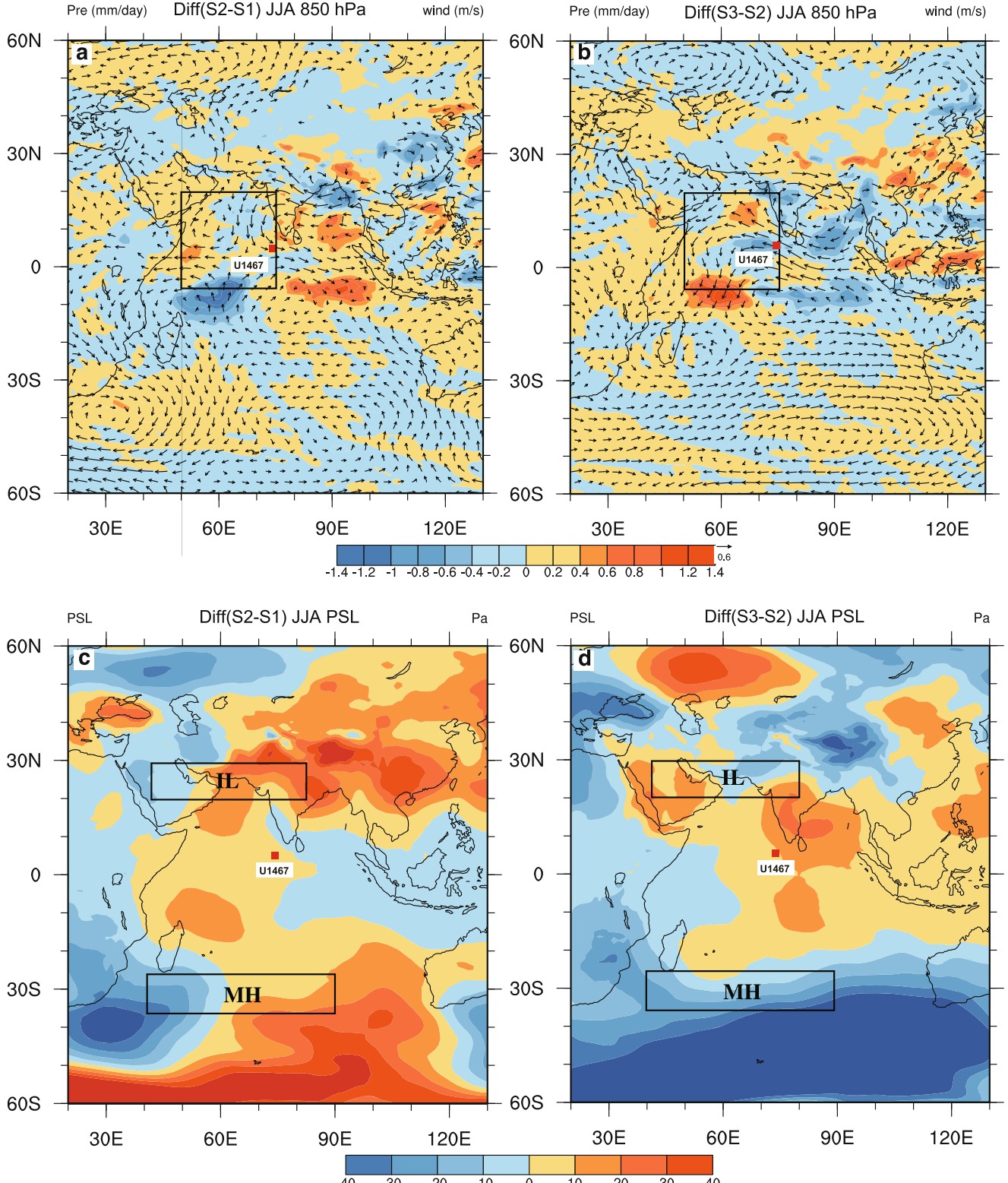

**Fig. 6 | Differences in precipitation (mm/day), the 850-hPa wind vectors (m/s) and the pressure of sea level (Pa) for the summer season (JJA) between different scenarios. a** Differences in precipitation and the 850-hPa wind vectors with the expansion of the West Antarctic ice-sheet (S2 minus S1). **b** Differences in precipitation and the 850-hPa wind vectors with the expansion of the Greenland ice sheet (S3 minus S2). **c** Differences in sea-level pressure with the expansion of the West Antarctic ice-sheet (S2 minus S1). **d** Differences in sea-level pressure with the expansion of the Greenland ice sheet (S3 minus S2). The rectangle denotes the area (50°–75°E, 5°S–20°N) selected for calculating the precipitation and the 850-hPa wind speed in Fig. 5a. The Mascarene High (MH; 40°–90°E, 25°–35°S) and the Indian Low (IL; 40°–80°E, 20°–30°N) are indicated and labeled. The red square refers to Site U1467.

yielded $^{87}$Sr/$^{86}$Sr $= 0.71032 \pm 0.000006$ (1σ; recommended value 0.71008–0.71060), and the JNdi-1 standard yielded $^{143}$Nd/$^{144}$Nd $= 0.512116 \pm 0.000005$ (1σ; recommended value 0.512108–0.512122). Nd results are expressed as εNd $= [(^{143}Nd/^{144}Nd_{(measured)}/^{143}Nd/^{144}Nd_{(CHUR)}) - 1] \times 10^4$, using the CHUR value of 0.512638 given by ref. [84].

## Sensitivity experiments

Sensitivity experiments were performed using the Community Earth System Model (CESM1.2) released by the National Center for Atmospheric Research (NCAR) in 2014. CESM1.2 is a fully coupled global climate model that consists of five components: atmosphere (the Community Climate System Model, CAM), land (the Community Land Model, CLM), ocean (the Parallel Ocean Program, POP), sea ice (the Sea-Ice Component, CICE) and the Coupler[85]. Early and the latest version of the model (CESM1.2) are often used to simulate past climate change[86], as well as present and future climate assessments[87] (IPCC AR5). In this study, the horizontal resolutions of the CAM and CLM are approximately 0.9° × 1.25° and the vertical dimensions contain 26 and 15 levels, respectively. The POP and CICE components have a horizontal resolution of 1° × 1° and depth (z) coordinates of 40 levels.

The response of the Mascarene High (MH), Indian Low (IL), MH-IL pressure gradient, summer (June, July, and August; JJA) precipitation, and 850-hPa wind speed were examined under three different scenarios (labeled S1, S2 and S3; Supplementary Fig. 3) which reflect increases in ice-sheet size for Antarctica and Greenland based on the literature[60,64–67]. S1 and S2 represent the ice cover accumulated in the East Antarctica and the whole Antarctica, respectively (Supplementary Fig. 3a, b). S3 represents the pre-industrial ice cover in Greenland superimposed on constant ice-sheet cover on the whole Antarctica (Supplementary Fig. 3c). In the sensitivity experiments, ice-sheet thickness variations are not considered. Hence, our simulated effects of the Antarctic and Greenland ice sheets on the SASM are entirely produced by changes in ice-induced albedo. The initial field and greenhouse gas concentration (e.g. $p$CO$_2$: 284 ppmv) for all the sensitivity experiments are based on the pre-industrial revolution scenario integrated from the model. The orbital parameters for 1950[88] are applied in the model simulation. To ensure that the output climate field of the numerical experiments has reached an equilibrium state, all of the numerical experiments are continuously integrated for 500 model years, and the average field for the last 50 years is used for analysis. Precipitation and wind speed are averaged for the region influenced by the SASM (50°–75°E, 5°S–20°N). The MH (40°–90°E, 25°–35°S) and IL (40°–80°E, 20°–30°N) cells are calculated from the atmospheric pressure at sea level averaged over the respective fixed areas. The MH-IL pressure gradient is calculated as the mean sea-level pressure over the Mascarene High and Indian Low pressure cell regions.

## Data availability

The strontium and neodymium isotopic composition data generated in this study are provided in the Supplementary Information.

## Code availability

The Community Earth System Model (CESM1.2) code can be accessed at https://www.cesm.ucar.edu/.

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

## Acknowledgements

We are grateful to the crew of the JOIDES Resolution and the shipboard scientific party for obtaining the samples during IODP Expedition 359. We thank the Ministry of Science and Technology (People's Republic of China) and the IODP-China for partially funding the expedition. We are grateful to Qiannan Hu helped with pretreatment of isotopes analysis and Xianwei Meng for discussion of manuscript. Z.Y., X.S, and Z.G. received funding from the National Natural Science Foundation of China (41776074, U1606401, 41690114), and the Taishan Scholar Program of Shandong (tspd20181216). X.L. acknowledges funding by the foundation of the Key Laboratory of Marine Geology and Metallogeny, MNR (MGM202001).

## Author contributions

Z.Y., X.S., and Z.G. conceived the paper. X.L. generated and analyzed the modeling data. Z.Y., B.N.N., C.B., S.L., and P.M. contributed the sediment provenance interpretation. H.Z. contributed to the analysis of Sr and Nd data. Z.Y. wrote the paper with contributions from all authors.

## Competing interests

The authors declare no competing interests.
