## [Peer Review File · Nature Communications]

Weakening of the South Asian summer monsoon linked to interhemispheric ice-sheet growth since 12 MaReviewer #1 (Remarks to the Author):

Key results: The manuscript titled "Two-step weakening of the South Asian summer monsoon linked to interhemispheric ice-sheets over the past 12 Myr" by Yao et al. contains twenty-nine new neodymium isotopes in the detrital fraction from the IODP Site U1467 and modeling results using the CESM1.2 model developed by the National Center for Atmospheric Research (NCAR) in 2014. The authors effectively partitioned ϵNd data into two groups: 0.06-4.8/6.58-12.35 Ma and 4.8 -6.58 Ma, which were used to highlight the proposed hypothesis. The authors also used other published data to provide further support for their hypothesis. Despite its novelty (i.e., new ϵNd data) and offering modeling results, the manuscript has fundamental shortcomings to be considered for a world-class and reputed publication like Nature Communications this time. I briefly outline my comments, critiques, and suggestions to improve it.

1. The authors should be commended for their efforts to produce new ϵNd data on the detrital fraction, as kinds of these data are seriously missing from the Indian Ocean. However, the manuscript does NOT address other factors that may or may not influence the ϵNd values in the detrital fractions, such as boundary exchange, ocean circulation, authigenic contribution, etc. The authors may wish to consider consulting publications such as Du et al. (2020 QSR), Lathika et al. (2021; QSR), Rashid et al. (2021), etc. that discuss issues with the ϵNd in the northern Indian Ocean and Jaume-Segui et al. (2021), etc. to gauge paradigm-shifting hypothesis in Nd geochemistry in marine sediments.

2. Diving deeper into the manuscript, it shows that four ϵNd data between 4.80 and 6.58 Ma abruptly diverge from the surprisingly monotonous trend in ϵNd values from 2.36 to 12.35 Ma if one detrends ϵNd data by calculating the mean and subtracting mean from each datum (Fig. 2). Further, the miss-match between ϵNd data at IODP Site U1447 and chlorite/smectite at the ODP Site 758 in the distal Bengal Fan and the overall decreasing trend in the smectite/(illite+chlorite) should have provided additional insights which the authors did not capitalize. For the record, some published ϵNd data for the same time interval from the ODP Site 758 (Gourlan et al., 2008; 2010) are available, which the authors may wish to consider when charting out the next course for their manuscript.

3. In addition to the new GEOTRACERS data, there are a lot of core top data available from the Arabian Sea that would help to fine-tune further their arguments that are lacking in the current manuscript. For example, Sirocko (1994; Abrupt change in monsoonal climate: evidence from the geochemical composition of Arabian Sea sediments) reported various LREE, including other bulk sediment geochemistry and isotopes from thirty-seven core tops. It would have been helpful to plot these data with those of the provenance isotopic data shown in Fig. 3 to assess the extent to which terrestrial ϵNd data match with those of the marine sediments. It would have lent credence or validated their technique to discriminate sediment sources. See also Dia et al. (1992, Marine Geology) for core top ϵNd data for the Indian Ocean.

4. The EICC and WICC stand for east Indian coastal current and west Indian coastal current, respectively, as such, these currents flow along the coast following the bathymetric constriction. In contrast to the EICC and WICC, the authors correctly pointed out that the SMC and WMC stand for summer monsoon current and winter monsoon current, respectively. The WICC or EICC would not deflect away from the coast while carrying buoyant freshwater due to the Coriolis force and rotation of the Earth. Is there a possibility that the authors may have confused these two sets of currents in explaining the Ganges-Brahmaputra transport to the IODP Site U1467? Given the deep bathymetric separation (see the figure below) between India and Chagos-Laccadive Plateau, it would be an unrealistic proposition to receive any detrital or other contribution from the Ganges-Brahmaputra discharge via EICC or WICC at the Site U1467.

5. I wonder whether the authors explored the possibility of any contribution by the

Antarctic Intermediate Water (AAIW) borne Nd to the IODP Site 1467, given its location at a shallow water depth of 487 m?

6. I find the paragraphs between lines 224 and 297 are a mix between facts and fiction. The authors suggested that “the changes in the MH and IL pressure system, meridional temperature gradients, and the polar ice-sheet growth were linked coherently, which generated the two-step decline of the SASM since ~12 Ma (lines 237-240)”. I had to jog my memory about Zachos contributions to polar ice-sheets growth during the Cenozoic. The Northern Hemisphere’s ice sheets began to appear ~8 Ma (Science 292, 2001; Fig. 2), and the Antarctic Ice Sheets (i.e., East and West) were in their place well before ~8 Ma. Therefore, it would have been better if the authors toned down their claim a bit and used the existing literature to further their hypothesis.

Validity: It is unclear whether the authors validated eNd used in the manuscript as the issues of boundary exchange, fossil fish teeth/debris, and detrital fraction to validate authigenic versus detrital ϵ Nd data at the IODP Site U1467. The authors appear to either ignore or unaware of these factors that may influence eNd data. For a quick review of these issues, I would like to suggest that the author consult publications such as Du et al. (2020 QSR), Lathika et al. (2021), Gourelan et al. (2008), Rashid et al. (2021), and so on. Note that this literature documents exclusively the eNd data from the northern Indian Ocean and their connection to the SASM and the Indian Ocean circulation.

Significance: These are the first set of data in the detrital fraction from a “relatively” older stratigraphy (i.e., Miocene) compared to the abundant data for the late Pleistocene. However, Yao et al. appear to set their sight to explain the eNd data as a function of the strength of the SASM without providing any discussion or discounting the impact of the Indian Ocean intermediate and deep circulation and boundary exchange. I would extend my neck and say that there has been a revolution in interpreting eNd data over the last three years compared to traditional views using the eNd data. Therefore, it is expected that the authors would provide multiple hypotheses and then settle on one that supports their data. That scientific thinking or rigor appears to be absent in the manuscript.

Data and Methodology: The authors used an established and time-tested methodology to acquire ϵ Nd data from the IODP Site U1467. I am not a modeler; however, I am familiar with and have been using model results obtained by the earlier version of the Community Earth System Model.

Analytical approach: The authors applied acceptable standard international analytical protocol used to acquire ϵ Nd data in the manuscript currently used by the Nd community.

Clarity and context: The manuscript text must be contextualized (see above) and improved the text.

References: I would like to suggest that the authors consider improving the list of references by following: (1) first cite references available from the northern Indian Ocean which has been inadequately referenced; (2) reduce the superfluous references that have no bearing to the topic of discussion, and (3) add or cite a few references that have divergent views (it does not mean that the authors need to accept those views).

Your expertise: I am comfortable assessing the content of the manuscript with ease as I am familiar with the ϵ Nd data through my publications from the northern Indian and the North Atlantic oceans. I have also provided reviews of manuscripts from numerous journals and research proposals dealing with the Nd geochemistry from the North Atlantic and Indian oceans.

Reviewer #1 Attachment on the following page

Redacted

Reviewer #2 (Remarks to the Author):

This paper presents a geological record and a modelling approach to provide an insight about the mechanisms and causes behind the evolution of the South Asian Summer Monsoon over the last 12 million years. The rationale for the study is that the SASM has been understudied compared to the East Asian Monsoon due to a scarcity of suitable records. However, it is, today, an important climatic system that affects the wellbeing of billions of people. We are here presented with a multi proxy record, using strontium and neodymium isotopic compositions of the sediments collected at IODP Exp. 359 Site U1467 in the Maldives Inner Sea. The paradigm here is that the source of terrigenous components in the ocean can be traced using Sr and Nd isotope ratios. In addition, variation in the $^{87}\text{Sr}/^{86}\text{Sr}$ ratio can be used to reflect the degree of weathering on the continent, therefore a climatic signal. Based on previous studies, the authors have defined different sources of the terrigenous component based on the ϵNd versus $^{87}\text{Sr}/^{86}\text{Sr}$. Data from site U1467 are compared to other records which contain proxies as indicators of terrigenous sources (chlorite/smectite).

The second component of the paper is a modelling approach of climatic conditions under different scenarios in relation to ice-sheet development in the high latitudes of both hemispheres.

The strength of the study is a methodological approach that is robust and a dataset showing changes concomitant with a global cooling signal, highlighting the weakening of the SASM over the last 12 million years, including an abrupt decline around 7 to 5 Ma. The mechanisms to explain the overall decline and abrupt decrease are also effectively discussed using the reconstructed wind speed, precipitation and sea level pressure scenarios.

However, there are some weaknesses in this paper that need to be addressed.

Firstly, why not analyse the chlorite/smectite composition in site U1467 so comparison between proxies would be stronger; in this paper, cores ODP658 and U1447 are in a different basin compared to U1467.

Secondly, the structure of the paper is not adequate in place. The location of the core is rather important to understand why the sediments there can be used to study eolian components. I would suggest moving the text from lines 155 to 175 to before line 103. Figure 3 should be presented before figure 2a, that means to move lines 79-86 after line 133.

Below are some suggestions corrections:

Line 54: add "and wellbeing" after "economic"

Line 95: correct "was" with "were"

Line 106: replace B-G with G-B

Line 126: replace river with River

Line 156 replace enclose with encloses

Line 156 replace is with are

Line 188: Add "a" after as

Figure 1 caption is incomplete, add d after the c in line 601

Figure 2: redraw your ϵNd using a smaller X-axis scale (from -8 to -13.5).

Line 649: correct deonotes with denotes

Figure 6: the red square should also be in maps c and d.

Reply to the comments on “Two-step weakening of the South Asian summer monsoon linked to interhemispheric ice-sheet growth over the past 12 Myr” by Yao et al.

Reply to the comments from reviewer #1

General comment:

The manuscript titled “Two-step weakening of the South Asian summer monsoon linked to interhemispheric ice-sheets over the past 12 Myr” by Yao et al. contains twenty-nine new neodymium isotopes in the detrital fraction from the IODP Site U1467 and modeling results using the CESM1.2 model developed by the National Center for Atmospheric Research (NCAR) in 2014. The authors effectively partitioned ϵNd data into two groups: 0.06-4.8/6.58-12.35 Ma and 4.8-6.58 Ma, which were used to highlight the proposed hypothesis. The authors also used other published data to provide further support for their hypothesis. Despite its novelty (i.e., new ϵNd data) and offering modeling results, the manuscript has fundamental shortcomings to be considered for a world-class and reputed publication like Nature Communications this time. I briefly outline my comments, critiques, and suggestions to improve it.

Comment 1: The authors should be commended for their efforts to produce new ϵNd data on the detrital fraction, as kinds of these data are seriously missing from the Indian Ocean. However, the manuscript does NOT address other factors that may or may not influence the ϵNd values in the detrital fractions, such as boundary exchange, ocean circulation, authigenic contribution, etc. The authors may wish to consider consulting publications such as Du et al. (2020 QSR), Lathika et al. (2021; QSR), Rashid et al. (2021), etc. that discuss issues with the ϵNd in the northern Indian Ocean and Jaume-Segui et al. (2021), etc. to gauge paradigm-shifting hypothesis in Nd geochemistry in marine sediments.

Reply: Many thanks for the valuable suggestions. Nearly all the papers suggested relate to water column/dissolved Nd inferred from selectively extracted authigenic fraction. We have, however, consulted these papers and also several other papers related to Nd isotopic study in the marine sediments in the Indian Ocean, and have cited some of the most relevant publications in the text. By citing these papers, we have discussed the potential influence of boundary exchange, ocean circulation (e.g. AAIW) borne Nd on our sediment Nd isotope data, and concluded that these processes have no pronounced influence on the ϵNd of the detrital fraction at our studied site (Site U1467).

Accordingly, the following text has been added in the revised manuscript (*P5, Line 144-161*).

“It has been highlighted that the Nd cycle in the ocean is complicated and could be influenced by factors such as fluvial input, boundary exchange, and ocean-circulation borne Nd³⁷⁻⁴⁰. Boundary exchange has been used mainly to explain the effect of sediments on dissolved Nd in seawater and not vice versa⁴¹, since the Nd concentration in sediments is 2-3 orders of magnitude higher than in dissolved seawater. Moreover, the lithic grains in the study area were primarily of eolian origin with relatively slow accumulation rates, and thus were less influenced by the boundary exchange which is often significant in the continental margins with high sediments supply^{37,41}. Therefore, we could preclude a significant influence of these processes on detrital Nd values at Site U1467. Although the Antarctic Intermediate Water (AAIW) has potentially influenced the intermediate water of the northern Indian Ocean⁴², but the ϵNd of AAIW is generally fluctuated between -8.4 to -6 for the glacial and interglacial periods^{43,44}, which is far more radiogenic than those of our lithogenic fraction sediment data. Further, our detrital Nd record is quite different from the seawater Nd pattern of intermediate waters of ODP Sites 707 (1552 m) and 775 (1661 m) in the northern Indian Ocean⁴⁵ (Supplementary Fig. 3). All the above evidences suggest that the authigenic Nd originated from the AAIW has no pronounced influence on the ϵNd of the detrital fraction at Site U1467.”

Supplementary Fig. 3. Comparison of detrital ϵNd at ODP Site U1467 with seawater ϵNd at ODP Sites 757 and 707 in the northern Indian Ocean⁴⁵. The ϵNd ranges of the glacial and modern AAIW^{43,44} are indicated by blue and red rectangles.

Comment 2: Diving deeper into the manuscript, it shows that four ϵNd data between 4.80 and 6.58 Ma abruptly diverge from the surprisingly monotonous trend in ϵNd values from 2.36 to 12.35 Ma if one detrends ϵNd data by calculating the mean and subtracting mean from each datum (Fig. 2). Further, the miss-match between ϵNd data at IODP Site U1447 and chlorite/smectite at the ODP Site 758 in the distal Bengal Fan and the overall decreasing trend in the smectite/(illite+chlorite) should have provided additional insights which the authors did not capitalize. For the record, some published ϵNd data for the same time interval from the ODP Site 758 (Gourlan et al., 2008; 2010) are available, which the authors may wish to consider when charting out the next course for their manuscript.

Reply: It is indeed that the ϵNd between 4.80 and 6.58 Ma are fairly less radiogenic. We interpret this pattern was caused by addition of materials with significant less radiogenic ϵNd (–20 to –18) from the southern Indian continent (Please see below for more detailed interpretation). We have re-examined the clay mineral data of IODP Site 1447 and ODP Site 578 in the northern Indian Ocean, we found that the ratio of kaolinite/chlorite could reflect the relative strength of the summer monsoon, as the kaolinite and smectite are mainly products of intense chemical weathering in warm humid climate, whereas the chlorite is related to the physical erosion under cold and dry climatic conditions. We thus have used this proxy to compare with our Nd record and these two records show largely consistent pattern (Fig. 3). We have also cited another long-term record of chemical weathering (CIA) from the northern South China Sea which also show fairly consistent change with our Nd record.

Different with kaolinite/chlorite in Site U1447 which is a proxy of summer monsoon, the ratio of chlorite/smectite at ODP Site 758 indicate an influence of the winter monsoon-related physical erosion in this region. The abrupt increase in chlorite/smectite at Site 758 implies a higher contribution of materials derived from the physical erosion of highlands of the Himalayan river basins, most likely transported by the East India Coastal Current under strong winter monsoonal wind. Thus the less radiogenic ϵNd values during the interval of ~7–5 Ma could be due to the combined effect of weekend summer monsoon which reduced the dust supply with more radiogenic Nd, and strengthened winter monsoon which transported more materials with significant less radiogenic ϵNd (–20 to –18) from the southern Indian continent.

As suggested by the reviewer, the seawater ϵNd data from the ODP Site 758 has been cited and compared with our detrital ϵNd record to clarify the influence of AAIW on our data. Please see the reply to comment 1.

All these corrections have been added to the revised manuscript (*P6, Line 194-200; P7, Line 208-221*).

“The CIA index at the ODP Site 1148 from the northern South China Sea also reveals similar monsoonal evolutionary pattern with our Nd records⁴⁹ (Fig. 3e). Changes in the kaolinite/chlorite ratio at Site U1447 in the western Andaman Sea, a proxy of chemical weathering in warm humid climate relative to the physical erosion under cold and dry climatic conditions⁵¹, demonstrate a weakening of the SASM since the Late Miocene (~ 10 Ma), interrupted by a further decrease at ~5-7 Ma⁵⁰ (Fig. 3c).”

“In addition to the influence of the weakened SASM which reduced the dust supply with more radiogenic Nd, the substantial decrease in the ϵNd values during the interval of ~7–5 Ma could be due to addition of sediment material with the least radiogenic ϵNd values (–20 to –18), possibly from the southern Indian continent¹⁹, to the core site (Fig. 2). These materials were possibly transported by the strengthened winter monsoonal winds, as previous studies have argued for a cold-drying climate and intensified winter monsoon during this time interval^{11,48,53}. Our interpretation is consistent with a two-fold increase in sediment accumulation rate at Site U1467 during the interval of ~7–5 Ma⁷ (Supplementary Fig. 1a). This inference is also supported by an abrupt increase in the ratio of chlorite/smectite from ODP Site 758 in the northeastern Indian Ocean (Fig. 3f), which implies a higher contribution of materials derived from the physical erosion of highlands of the Himalayan river basins⁵⁴, most likely transported by the East India Coastal Current⁵⁵ under strong winter monsoonal wind.”

Comment 3: In addition to the new GEOTRACERS data, there are a lot of core top data available from the Arabian Sea that would help to fine-tune further their arguments that are lacking in the current manuscript. For example, Sirocko (1994; Abrupt change in monsoonal climate: evidence from the geochemical composition of Arabian Sea sediments) reported various LREE, including other bulk sediment geochemistry and isotopes from thirty-seven core tops. It would have been helpful to plot these data with those of the provenance isotopic data shown in Fig. 3 to assess the extent to which terrestrial ϵNd data match with those of the marine sediments. It would have lent

credence or validated their technique to discriminate sediment sources. See also Dia et al. (1992, Marine Geology) for core top ϵNd data for the Indian Ocean.

Reply: Thanks for the very good suggestions that further validate our interpretation of provenance change. High-quality ϵNd dataset from core-top sediments in the Arabian Sea reported by Sirocko (1994)²⁹, and from marine sediments and rocks along the margins of the northeastern African and the southern Arabian Peninsula³³ have been cited to plot the provenance diagram (Fig. 2). We also generated a new figure (Supplementary Fig. 2) to show the spatial distribution of ϵNd in the northern Indian Ocean which display a clear progressively decreasing trend from the northeastern Africa/Arabian Peninsula to the western and eastern Arabian Sea^{29,33}. This changing pattern could provide support for our interpretation of provenance change at Site U1467. All these information has been added in the revised manuscript (*P4, Line 118-120; P5, Line 138-143*).

“The average ϵNd value of marine sediments and rocks from the margins of the northeastern African and the southern Arabian Peninsula too are highly radiogenic (average -4.6)³³.”

“Our predicted provenance change at Site U1467 is further supported by the Nd isotope of core-top sediments from the Arabian Sea and its surrounding margins^{29,33}. Spatial distribution of ϵNd values of the detrital fraction from these areas reveals clear progressively decreasing trend from the northeastern Africa/Arabian Peninsula to the western and eastern Arabian Sea^{29,33} (Supplementary Fig. 2).”

Supplementary Fig. 2. Spatial distribution of ϵNd values of the core-top sediments from the Arabian Sea (Squares)²⁹ and of the marine sediments/rocks along the Indian Ocean continental margin (dots)³³.

Comment 4: The EICC and WICC stand for east Indian coastal current and west Indian coastal current, respectively, as such, these currents flow along the coast following the bathymetric constriction. In contrast to the EICC and WICC, the authors correctly pointed out that the SMC and WMC stand for summer monsoon current and winter monsoon current, respectively. The WICC or EICC would not deflect away from the coast while carrying buoyant freshwater due to the Coriolis force and rotation of the Earth. Is there a possibility that the authors may have confused these two sets of currents in explaining the Ganges-Brahmaputra transport to the IODP Site U1467? Given the deep bathymetric separation (see the figure below) between India and Chagos-Laccadive Plateau, it would be an unrealistic proposition to receive any detrital or other contribution from the Ganges-Brahmaputra discharge via EICC or WICC at the Site U1467.

Reply: After examining the published provenance-related literature in this region and considering the bathymetry of studied area, we agree with the reviewer that the lithic sediments in this site were primarily of eolian origin and other detrital contribution from fluvial discharge is insignificant. Therefore, in the revised manuscript we have deleted the information related to the influence of fluvial input and the associated effects of circulation, and added more reference to support the interpretation of eolian origin.

Comment 5: I wonder whether the authors explored the possibility of any contribution by the Antarctic Intermediate Water (AAIW) borne Nd to the IODP Site 1467, given its location at a shallow water depth of 487 m?

Reply: Please refer to the reply to comment 1.

Comment 6: I find the paragraphs between lines 224 and 297 are a mix between facts and fiction. The authors suggested that “the changes in the MH and IL pressure system, meridional temperature gradients, and the polar ice-sheet growth were linked coherently, which generated the two-step decline of the SASM since ~12 Ma (lines 237-240)”. I had to jog my memory about Zachos contributions to polar ice-sheets growth during the Cenozoic. The Northern Hemisphere’s ice sheets began to appear ~8 Ma (Science 292, 2001; Fig. 2), and the Antarctic Ice Sheets (i.e., East and West) were in their place well before ~8 Ma. Therefore, it would have been better if the authors toned down their claim a bit and used the existing literature to further their hypothesis.

Reply: We have checked the publication of Zachos et al. (2001) and other published

papers. It is true that the Antarctic ice sheet expanded from ~13.9 Ma and reached the present condition by ~8 Ma. For the Northern Hemisphere glaciation history, there are different views that the ephemeral Northern Hemisphere ice sheet developed from 6–7 to 8 Ma. Therefore, we have rephrased these sentences in the revised manuscript (**P8, Line 252-256, Line 263-269**), and these corrections do not alter the main conclusion in the manuscript.

“There was a major expansion of the East Antarctic ice sheet from ~13.9 Ma onwards and a further expansion of the West Antarctic ice sheet until ~8 Ma^{60,64,65} (Fig. 4d) leading to faster cooling, and consequently to enhanced meridional temperature gradients, in the Southern Hemisphere relative to the Northern Hemisphere⁶² (Fig. 4b).

“The development of ephemeral Northern Hemisphere ice sheet between ~6-8 Ma^{60,66}, followed by the significant expansion of the Greenland ice sheet commenced at ~4 Ma^{60,66,67} (Fig. 4d), causing a cooling in the mid-latitude Northern Hemisphere and enhanced meridional temperature gradients⁶² (Fig. 4b). This pushes the Hadley Cell back southward, possibly leading to the subsiding and ascending branches of the Hadley Cell deviated from the MH and IL in the opposite direction.”

Comment 7: Validity: It is unclear whether the authors validated ϵNd used in the manuscript as the issues of boundary exchange, fossil fish teeth/debris, and detrital fraction to validate authigenic versus detrital ϵNd data at the IODP Site U1467. The authors appear to either ignore or unaware of these factors that may influence ϵNd data. For a quick review of these issues, I would like to suggest that the author consult publications such as Du et al. (2020 QSR), Lathika et al. (2021), Gourlan et al. (2008), Rashid et al. (2021), and so on. Note that this literature documents exclusively the ϵNd data from the northern Indian Ocean and their connection to the SASM and the Indian Ocean circulation.

Reply: Please see the reply to comment 1.

Comment 8: Significance: These are the first set of data in the detrital fraction from a “relatively” older stratigraphy (i.e., Miocene) compared to the abundant data for the late Pleistocene. However, Yao et al. appear to set their sight to explain the ϵNd data as a function of the strength of the SASM without providing any discussion or discounting the impact of the Indian Ocean intermediate and deep circulation and boundary

exchange. I would extend my neck and say that there has been a revolution in interpreting ϵNd data over the last three years compared to traditional views using the ϵNd data. Therefore, it is expected that the authors would provide multiple hypotheses and then settle on one that supports their data. That scientific thinking or rigor appears to be absent in the manuscript.

Reply: Please see the reply to comment 1.

Comment 9: Data and Methodology: The authors used an established and time-tested methodology to acquire ϵNd data from the IODP Site U1467. I am not a modeler; however, I am familiar with and have been using model results obtained by the earlier version of the Community Earth System Model.

Reply: Thank you.

Comment 10: Analytical approach: The authors applied acceptable standard international analytical protocol used to acquire ϵNd data in the manuscript currently used by the Nd community.

Reply: Thank you.

Comment 11: Clarity and context: The manuscript text must be contextualized (see above) and improved the text.

Reply: Done.

Comment 12: References: I would like to suggest that the authors consider improving the list of references by following: (1) first cite references available from the northern Indian Ocean which has been inadequately referenced; (2) reduce the superfluous references that have no bearing to the topic of discussion, and (3) add or cite a few references that have divergent views (it does not mean that the authors need to accept those views).

Reply: We have updated the reference list as suggested by the reviewer. Previously published Nd isotope papers on the Indian Ocean, as well as papers reviewing Nd cycling in the ocean have been cited, e.g. ref. 29, 37-45. We have also deleted some of references that are not essential to the discussed topic.

Comment 13: Your expertise: I am comfortable assessing the content of the manuscript with ease as I am familiar with the ϵNd data through my publications from the northern Indian and the North Atlantic oceans. I have also provided reviews of manuscripts from numerous journals and research proposals dealing with the Nd geochemistry from the North Atlantic and Indian oceans.

Reply: Done.

Reply to the comments from reviewer #2

General comment:

This paper presents a geological record and a modelling approach to provide an insight about the mechanisms and causes behind the evolution of the South Asian Summer Monsoon over the last 12 million years. The rationale for the study is that the SASM has been understudied compared to the East Asian Monsoon due to a scarcity of suitable records. However, it is, today, an important climatic system that affects the wellbeing of billions of people. We are here presented with a multi proxy record, using strontium and neodymium isotopic compositions of the sediments collected at IODP Exp. 359 Site U1467 in the Maldives Inner Sea. The paradigm here is that the source of terrigenous components in the ocean can be traced using Sr and Nd isotope ratios. In addition, variation in the $^{87}\text{Sr}/^{86}\text{Sr}$ ratio can be used to reflect the degree of weathering on the continent, therefore a climatic signal. Based on previous studies, the authors have defined different sources of the terrigenous component based on the ϵNd versus $^{87}\text{Sr}/^{86}\text{Sr}$. Data from site U1467 are compared to other records which contain proxies as indicators of terrigenous sources (chlorite/smectite). The second component of the paper is a modelling approach of climatic conditions under different scenarios in relation to ice-sheet development in the high latitudes of both hemispheres.

The strength of the study is a methodological approach that is robust and a dataset showing changes concomitant with a global cooling signal, highlighting the weakening of the SASM over the last 12 million years, including an abrupt decline around 7 to 5 Ma. The mechanisms to explain the overall decline and abrupt decrease are also effectively discussed using the reconstructed wind speed, precipitation and sea level pressure scenarios. However, there are some weaknesses in this paper that need to be addressed.

Comment 1: Firstly, why not analyze the chlorite/smectite composition in site U1467 so comparison between proxies would be stronger; in this paper, cores ODP758 and U1447 are in a different basin compared to U1467.

Reply: This is a very good suggestion. We have tried to extract the clay fraction (< 2 μm) in samples of Site U1467 to perform the XRD analysis. Unfortunately, we did not get sufficient clay fraction after removing the carbonates and organic matter to do this analysis, as the carbonates content of samples are generally higher than 80% and mostly above 90%. Although ODP Sites 758 and U1447 located in a different basin compared with our site, the clay minerals in these sites can still be used to reflect monsoon information within a large regional scale. Consequently, these sites have been used for comparison in the revised text. And a good match is seen. Thank you.

Comment 2: Secondly, the structure of the paper is not adequate in place. The location of the core is rather important to understand why the sediments there can be used to study eolian components. I would suggest moving the text from lines 155 to 175 to before line 103. Figure 3 should be presented before figure 2a, that means to move lines 79-86 after line 133.

Reply: We totally agree with the reviewer, and have adjusted the order of these two paragraphs, and consequently the Figures in the revised manuscript (changed the order of Fig.2 and Fig. 3).

Other comments:

Below are some suggestions corrections:

Line 54: add “and wellbeing” after “economic”

Line 95: correct “was” with “were”

Line106: replace B-G with G-B

Line 126: replace river with River

Line 156 replace enclose with encloses

Line 156 replace is with are

Line 188: Add “a” after as

Figure 1 caption is incomplete, add d after the c in line 601

Figure 2: redraw your ϵNd using a smaller X-axis scale (from -8 to -13.5).

Line 649: correct deonotes with denotes

Figure 6: the red square should also be in maps c and d.

Reply: We are grateful to the reviewer pointing out these errors. We have corrected all of them and checked the grammar throughout the text carefully.

References

7. Betzler, C. et al. The abrupt onset of the modern South Asian Monsoon winds. *Sci. Rep.* 6, 1–10 (2016).
11. Holbourn, A. E. et al. Late Miocene climate cooling and intensification of southeast Asian winter monsoon. *Nat. Commun.* 9, 1584 (2018).
19. Kessarkar, P. M., Rao, V. P., Ahmad, S. M. & Babu, G. A. Clay minerals and Sr-Nd isotopes of the sediments along the western margin of India and their implication for sediment provenance. *Mar. Geol.* 202, 55–69 (2003).
29. Sirocko, F. Abrupt change in monsoonal climate: evidence from the geochemical composition of Arabian Sea sediments. Habilitation Thesis, 216 pp., University of Kiel (1994).
33. Jeandel, C., Arsouze, T., Lacan, F., Téchiné, P. & Dutay, J. C. Isotopic Nd compositions and concentrations of the lithogenic inputs into the ocean: A compilation, with an emphasis on the margins. *Chem. Geol.* 239, 156–164 (2007).
37. Lacan, F. & Jeandel, C. Neodymium isotopes as a new tool for quantifying exchange fluxes at the continent-ocean interface. *Earth Planet. Sci. Lett.* 232, 245–257 (2005).
38. Du, J., Haley, B. A. & Mix, A. C. Evolution of the Global Overturning Circulation since the Last Glacial Maximum based on marine authigenic neodymium isotopes. *Quat. Sci. Rev.* 241, 106396 (2020).
39. Rashid, H., Wang, Y. & Gurlan, A. T. Impact of climate change on past Indian monsoon and circulation: A perspective based on radiogenic and trace metal geochemistry. *Atmosphere* 12, 330 (2021).
40. Lathika, N. et al. Deep water circulation in the Arabian Sea during the last glacial cycle: Implications for paleo-redox condition, carbon sink and atmospheric CO₂ variability. *Quat. Sci. Rev.* 257, 106853 (2021).
41. Jeandel, C. Overview of the mechanisms that could explain the ‘Boundary

- exchange' at the Land–Ocean contact. *Proc. Math. Phys. Eng. Sci.* 374, 1–13 (2016).
42. Yu, Z. et al. Antarctic Intermediate Water penetration into the Northern Indian Ocean during the last deglaciation. *Earth Planet. Sci. Lett.* 500, 67–75 (2018).
 43. Howe, J. N. et al. North Atlantic deep water production during the last glacial maximum. *Nat. Commun.* 7, 11765 (2016).
 44. Stichel, T., Frank, M., Rickli, J. & Haley, B. A. The hafnium and neodymium isotope composition of seawater in the Atlantic sector of the Southern Ocean. *Earth Planet. Sci. Lett.* 317, 282–294 (2012).
 45. Gourlan, A.T., Meynadier, L. & Allègre, C. J. Tectonically driven changes in the Indian Ocean circulation over the last 25 Ma: Neodymium isotope evidence. *Earth Planet. Sci. Lett.* 267, 353–364 (2008).
 48. Clift, P. D. et al. Correlation of Himalayan exhumation rates and Asian monsoon intensity. *Nat. Geosci.* 1, 875–880 (2008).
 49. Wei, G., Li, X. H., Liu, Y., Shao, L. & Liang, X. Geochemical record of chemical weathering and monsoon climate change since the early Miocene in the South China Sea. *Paleoceanography* 21, 1-11 (2006).
 50. Lee, J. et al. Monsoon-influenced variation of clay mineral compositions and detrital Nd-Sr isotopes in the western Andaman Sea (IODP Site U1447) since the late Miocene. *Palaeogeogr. Palaeoclimatol. Palaeoecol.* 538, 109339 (2020).
 51. Chamley, H. *Clay Sedimentology*. Berlin, Germany, Springer-Verlag, pp267 (1989).
 53. Li, F. J., Rousseau, D. D., Wu, N. Q., Hao, Q. Z. & Pei, Y. P. Late Neogene evolution of the East Asian monsoon revealed by terrestrial mollusk record in Western Chinese Loess Plateau: from winter to summer dominated sub-regime. *Earth Planet. Sc. Lett.* 274, 439–447 (2008).
 54. Song, Z. et al. Paleoenvironmental evolution of South Asia and its link to Himalayan uplift and climatic change since the late Eocene. *Glob. Planet. Change* 200, 103459 (2021).
 55. Shankar, D., Vinayachandran, P. N. & Unnikrishnan, A. S. The monsoon currents in the north Indian Ocean. *Prog. Oceanogr.* 52, 63–120 (2002).
 62. Herbert, T. D. et al. Late Miocene global cooling and the rise of modern ecosystems. *Nat. Geosci.* 9, 843–847 (2016).
 60. Zachos, J., Pagani, M., Sloan, L., Thomas, E. & Billups, K. Trends, Rhythms, and

- aberration in global climate 65 Ma to present. *Science* 292, 686–693 (2001).
64. Kennett, J. P. & Barker, P. F. Latest Cretaceous to Cenozoic climate and oceanographic developments in the Weddell Sea, Antarctica: an ocean-drilling perspective. *Proc. Ocean Drill. Prog. Sci. Results*, 937–960 (1990).
 65. Holbourn, A., Kuhnt, W., Schulz, M. & Erlenkeuser, H. Impacts of orbital forcing and atmospheric carbon dioxide on Miocene ice-sheet expansion. *Nature* 438, 483–487 (2005).
 66. Larsen, H. C. et al. Seven million years of glaciation in Greenland. *Science* 264, 952–955 (1994).
 67. Maslin, M. A., Li, X. S., Loutre, M. F. & Berger, A. The contribution of orbital forcing to the progressive intensification of Northern Hemisphere glaciation. *Quat. Sci. Rev.* 17, 411–426 (1998).

Reviewer #2 (Remarks to the Author):

I am satisfied with the responses to my comments and the changes made to the ms. I am happy to accept this revised version.

Reviewer #3 (Remarks to the Author):

Re-review of the manuscript titled "Two-step weakening of the South Asian summer monsoon linked to interhemispheric ice-sheets over the past 12 Myr" by Yao et al. The authors have successfully addressed my suggestions, comments, and critiques of their manuscript. I am satisfied with the authors' effort to incorporate many available published data and information in the literature to further firm up their scientific interpretation. However, I have a few minor suggestions/comments which may additionally provide clarity and improve the presentation of their findings, as given below.

1. I wonder whether the authors could "somehow" plot the seawater Nd data currently illustrated in Supplementary Fig. 3. These data could be plotted in one of the figures, perhaps, in Figure 4. At present, it is "hidden" as a supplementary figure, and most readers would not bother to browse it or benefit from the hard work of the authors.

2. Supplementary Fig. 2. I wonder whether the authors would consider plotting the eNd value rather than plotting the core-tops position and marine sediments' location. The lack of these data prevents us from further assessing the nature of provenances and their respective eNd values. It would also be worthwhile to submit these data as an xls file (as a supplementary file) with the latitude, longitude, water depth, eNd, etc.

3. The manuscript could use some professional editing to remove many "Introductory sentences that appear like "Geology 101,". For example, "Lines 79-86", 224-232, etc. Further, Lines 36-37 and 52-54 appear identical; there is no linkage between lines 36-37 and 38-41, and so on.

4. The authors could use "geological editing" as many "unscientific terms" are prevalent in it. For example, "riverine materials," "eolian materials," "By contrast, the Sr and Nd isotopic composition of the Indian Thar deserts "sands" showed, "sands by definition are SiO₂ and do NOT have any isotopic composition". I could go on, but I made my point here.

Reply to the comments on “Two-step weakening of the South Asian summer monsoon linked to interhemispheric ice-sheet growth over the past 12 Myr” by Yao et al.

Reply to the comments from reviewer #3

General comment:

Re-review of the manuscript titled “Two-step weakening of the South Asian summer monsoon linked to interhemispheric ice-sheets over the past 12 Myr” by Yao et al. The authors have successfully addressed my suggestions, comments, and critiques of their manuscript. I am satisfied with the authors’ effort to incorporate many available published data and information in the literature to further firm up their scientific interpretation. However, I have a few minor suggestions/comments which may additionally provide clarity and improve the presentation of their findings, as given below.

Comment 1: I wonder whether the authors could “somehow” plot the seawater Nd data currently illustrated in Supplementary Fig. 3. These data could be plotted in one of the figures, perhaps, in Figure 4. At present, it is “hidden” as a supplementary figure, and most readers would not bother to browse it or benefit from the hard work of the authors.

Reply: As suggested by the reviewer, we move the information of Supplementary Fig. 3 to Figure 4.

Figure 4. Comparisons of the South Asian summer monsoon (SASM) evolution with other climatic records. **a** Comparison of detrital ϵNd (2sigma S.D.) at Site U1467 with seawater ϵNd at ODP Sites 757 and 707 in the northern Indian Ocean⁴⁵. The ϵNd ranges of the glacial and modern Antarctic Intermediate Water (AAIW)^{43,44} are indicated by blue and red rectangles. **b** Stacks of alkenone $U^{k'_{37}}$ -based sea-surface temperature (SST) anomalies relative to today for subtropical (30-50° N, 30-50° S) and tropical regions⁶², with lower values indicating strong meridional temperature gradients (MTG). **c** Meridional temperature gradients of the Pacific calculated between sites from the northern subtropical and equator region⁶¹. **d** Benthic oxygen isotope record from ODP Site 1146¹¹ (21-point moving average) along with blue and green bars denoting the generally accepted chronology of southern and Northern Hemisphere Cenozoic glaciation^{60,64-67}. The vertical shaded area represents the time interval of ~7–5 Ma.

Comment 2: Supplementary Fig. 2. I wonder whether the authors would consider plotting the ϵNd value rather than plotting the core-tops position and marine sediments'

location. The lack of these data prevents us from further assessing the nature of provenances and their respective ϵNd values. It would also be worthwhile to submit these data as an xls file (as a supplementary file) with the latitude, longitude, water depth, ϵNd , etc.

Reply: As suggested by the reviewer, we also plot the ϵNd value along a W-E direction within 0-30° N regions in this Figure. As these data were not generated by this study and can be easily accessed from ref 29 and ref 33, these data thus were not included in the supplementary file.

Supplementary Fig. 2. Changes in the ϵNd values of the core-tope sediments from the Indian Ocean margin. Spatial distribution of ϵNd values of the core-tope sediments from the Arabian Sea (Squares)²⁹ and of the marine sediments/rocks along the Indian Ocean continental margin (dots)³³. The ϵNd values within the regions of 0-30° N are also shown in the bottom panel.

Comment 3: The manuscript could use some professional editing to remove many “Introductory sentences that appear like “Geology 101,”. For example, “Lines 79-86”, 224-232, etc. Further, Lines 36-37 and 52-54 appear identical; there is no linkage between lines 36-37 and 38-41, and so on.

Comment 4: The authors could use “geological editing” as many “unscientific terms” are prevalent in it. For example, “riverine materials,” “eolian materials,” “By contrast, the Sr and Nd isotopic composition of the Indian Thar deserts “sands” showed, “sands by definition are SiO₂ and do NOT have any isotopic composition”. I could go on, but

I made my point here.

Reply: We have read through the manuscript carefully and have corrected the inappropriate expression and removed the redundant sentences.